# The Crucial Role of Inflammation and the Immune System in Colorectal Cancer Carcinogenesis: A Comprehensive Perspective

**DOI:** 10.3390/ijms25116188

**Published:** 2024-06-04

**Authors:** Antonio Manuel Burgos-Molina, Teresa Téllez Santana, Maximino Redondo, María José Bravo Romero

**Affiliations:** 1Surgery, Biochemistry and Immunology Department, School of Medicine, University of Malaga, 29010 Málaga, Spain; aburgos@uma.es (A.M.B.-M.); teresatellez@uma.es (T.T.S.); mjbravo@uma.es (M.J.B.R.); 2Research Network on Chronic Diseases, Primary Care, and Health Promotion (RICAPPS), Carlos III Health Institute (Instituto de Salud Carlos III), Av. de Monforte de Lemos, 5, 28029 Madrid, Spain; 3Málaga Biomedical Research Institute (Instituto de Investigación Biomédica de Málaga, IBIMA), Calle Doctor Miguel Díaz Recio, 28, 29010 Málaga, Spain; 4Research Unit, Hospital Costa del Sol, Autovía A-7, km 187, 29603 Marbella, Spain

**Keywords:** inflammation, colorectal cancer, carcinogenesis, microbiota, microbiome

## Abstract

Chronic inflammation drives the growth of colorectal cancer through the dysregulation of molecular pathways within the immune system. Infiltration of immune cells, such as macrophages, into tumoral regions results in the release of proinflammatory cytokines (IL-6; IL-17; TNF-α), fostering tumor proliferation, survival, and invasion. Tumors employ various mechanisms to evade immune surveillance, effectively ‘cloaking’ themselves from detection and subsequent attack. A comprehensive understanding of these intricate molecular interactions is paramount for advancing novel strategies aimed at modulating the immune response against cancer.

## 1. Introduction

Colorectal cancer is a widespread type of cancer that affects the colon and rectum, posing a significant public health challenge. It ranks among the most common cancers globally, with more than 1.8 million new cases diagnosed each year and approximately 900,000 deaths [1]. It is the second most frequently diagnosed cancer in women and the third most frequently diagnosed in men [2,3,4]. Colorectal cancer associated with colitis (CAC) constitutes about 5% of all cases of colorectal cancer [5]. CAC develops from persistent inflammation, especially in the context of inflammatory bowel disease (IBD), which has been steadily increasing in both incidence and prevalence over time [6,7]. In general, the incidence of IBD has increased in recent decades globally, affecting between 5 and 15 million people worldwide. Incidence rates are higher in developed countries, especially in northern Europe and North America, but data suggest that incidence is also increasing in developing countries. Specifically, the incidence of Crohn’s disease is 5 to 15 cases per 100,000 population per year, while the incidence of ulcerative colitis is 5 to 12 cases per 100,000 population per year. Regarding chronic inflammatory diseases, such as ulcerative colitis and Crohn’s disease (CD), also known as IBD, they have experienced an increase in incidence worldwide. While incidence remains higher in North America and Europe, epidemiological data indicate an increase in regions such as Asia, the Middle East, and Latin America. These chronic inflammatory diseases not only increase the risk of CRC, but their increasing incidence suggests a relationship with the observed rise in cases of colorectal cancer in young adults [8]. 

Most cases of colorectal cancer (CRC) are associated with environmental factors rather than any specific heritable genes [9] Risk factors include exposure to environmental and foodborne mutagens, certain intestinal bacteria, and pathogens, as well as chronic inflammation of the intestine, which precedes the development of tumors.

## 2. Colorectal Cancer and Intestinal Disease

Carcinogenesis, a complex process influenced by genetic, environmental, and epigenetic factors, involves the uncontrolled proliferation of abnormal cells, which characterizes cancer. In this context, chronic inflammation emerges as a crucial factor. This prolonged immune response, mediated by various cells and molecules, such as macrophages, lymphocytes, and proinflammatory cytokines, such as tumor necrosis factor (TNF)-α, interleukin (IL)-6, and IL-1β, not only promotes tumor progression but also modifies the tumor microenvironment (TME) in several ways. In addition to activating molecular pathways, such as NF-κB and MAPK, chronic inflammation may predispose individuals to cancer development by promoting the accumulation of genetic mutations and the selection of more aggressive cellular variants.

Once the tumor is established, it can perpetuate chronic inflammation through the release of proinflammatory factors and the recruitment of suppressive immune cells, further altering the TME. This process also suppresses the antitumor immune response by recruiting suppressive immune cells and inhibiting antigen presentation and activation of cytotoxic T cells, favoring cancer progression. Therefore, the interaction between chronic inflammation and the tumor environment is bidirectional and plays a fundamental role in carcinogenesis and cancer progression [10]. 

It is widely recognized today that patients with IBD, especially ulcerative colitis (UC) and, to a lesser extent, Crohn’s disease (CD), have a higher risk of colorectal cancer. This type of cancer, which arises in the context of IBD, is considered a representative example of how inflammation can trigger cancer development. Chronic inflammation results in DNA damage due to oxidative stress, activating genes that promote tumor growth and deactivating genes that inhibit it [11]. In addition, elements generated by the body’s immune response, with contributions from the intestinal microbiome and its products, also participate in this inflammatory and carcinogenic process, leading to a sequence of events that includes genetic and epigenetic changes, followed by clonal expansion of somatic epithelial cells, influenced by the immunological environment and surrounding tissue. This inflammation can increase the risk of cancer by providing bioactive molecules, such as cytokines, growth factors, and chemokines, which maintain a sustained proliferative rate and cell survival signals and promote the formation of new blood vessels. As a result, these substances disrupt metabolic pathways, increase inflammation levels, and induce mutations in genes associated with cancer. This leads to the development of dysplasia in the epithelial cells of the intestine, stimulating cell growth and various forms of cell death, thereby worsening the disease. Once the immunosuppressive TME is established, tumor cells can evade detection by the immune system and adapt their energy metabolism to cope with low glucose levels and the accumulation of metabolic waste. Additionally, enzymes that modify the extracellular matrix, such as metalloproteinases, facilitate processes like epithelial–mesenchymal transition (EMT) and other carcinogenic programs, including genome instability, reprogramming of energetic metabolism, and evasion of the immune system [12,13,14].

Colorectal cancer (CRC) and inflammatory bowel disease (IBD) are closely linked conditions, with significant implications for patient health and treatment.

## 3. Inflammation Induced by Reactive Oxygen Species (ROS) and Reactive Nitrogen Species (RNS) in Carcinogenesis

In an inflammatory response, the activation of epithelial and immune cells triggers the generation of ROS and RNS. This occurs through the induction of two key enzymes: nicotinamide adenine dinucleotide phosphate (NADPH) oxidase and nitric oxide synthase (NOS).

NADPH oxidase is a protein complex that catalyzes the production of the superoxide anion (O_2_^−^), which in turn leads to the formation of hydrogen peroxide (H_2_O_2_) through the action of superoxide dismutase (SOD). NADPH oxidase is expressed in both phagocytic and non-phagocytic cells, and there are different isoforms of cytochrome subunits in various cell types (e.g., NOX_2_ in phagocytic cells, such as macrophages and neutrophils, and NOX_1_, 3–5 and DUOX1, 2 in non-phagocytic cells) [15].

On the other hand, NOS generates nitric oxide (NO) from L-arginine, which can be converted into RNS, such as nitrogen dioxide (NO_2_∙), peroxynitrite (ONOO^−^), and dinitrogen trioxide (N_2_O_3_). Different NOS isoforms are expressed according to the cell type: inducible NOS (iNOS) in phagocytic cells and constitutive (eNOS and nNOS) in endothelial and neuronal cells [16]. Additionally, certain cytokines, such as TNF-α, IL-6, and TGF-β, also trigger RONS (reactive oxygen and nitrogen species) generation in non-phagocytic cells [17,18].

The activation of phagocytic cells can induce the production of RONS, while TNF-α, IL-6, and TGF-β can trigger their generation in non-phagocytic cells [19,20]. An increase in the expression of NADPH oxidase and NOS and their RONS products has been observed in various types of cancer and chronic inflammatory diseases, such as *H. pylori*-associated gastritis and inflammatory bowel diseases (IBD), suggesting a role in genesis and malignant progression [21]. 

Specifically in colorectal cancer, RONS play a crucial role in the carcinogenesis of this pathology through mechanisms involving DNA damage, oxidative stress, and inflammation specific to the colorectal environment. Both ROS and RNS can cause direct DNA damage. In the case of ROS (superoxide (O_2_^−^) and hydrogen peroxide (H_2_O_2_)), they can induce damage by oxidizing DNA bases, causing strand breaks and adduct formation, which can interfere with DNA replication and transcription, leading to oncogenic and tumor suppressor mutations that promote carcinogenesis [22].

As for RNS, nitric oxide (NO) can react with superoxide to form peroxynitrite (ONOO^−^), a reactive species that can nitrate DNA bases and cause oxidative deamination, resulting in mutations. Peroxynitrite (ONOO^−^) can induce guanine nitration, forming 8-nitroguanine, which contributes to mutagenesis and genomic instability [23]. 

One of the primary types of DNA damage is the formation of 8-oxo-7,8-dihydro-2′-deoxyguanosine (8-oxodG), which is used as a marker of oxidative DNA damage. This compound is produced when guanine, which is particularly vulnerable due to its low ionization energy, is oxidized. 8-oxodG is problematic because it can mispair with adenine during DNA replication, leading to mutations. Additionally, this compound can interfere with DNA repair mechanisms. Several studies have found that deficient activity of DNA repair proteins is related to enzymatic S-nitrosylation, a process intensified by increased RNS levels [24].

ROS can generate oxidative stress in colon cells, resulting in an imbalance between the production of reactive species and the body’s antioxidant capacity. This oxidative stress can cause cellular damage, disrupting the normal functions of colon cells and promoting uncontrolled cell proliferation. Moreover, it can activate and regulate cellular signaling pathways that promote tumor survival and growth [25].

This is the case for the Akt, Erk1/2, and HIF-1 pathways, which are involved in regulating cell proliferation, survival, angiogenesis, and adaptation to the cellular environment. Activation of Akt and Erk1/2 leads to cell proliferation through two main mechanisms. Firstly, it alters the activity of certain proteins that have cysteine residues susceptible to oxidation. Secondly, it stimulates the expression of genes related to cell proliferation, such as c-Myc and Cyclin D1, facilitating cell cycle progression and tumor cell multiplication [26,27].

On the other hand, Erk1/2 activation can also induce the expression of anti-apoptotic proteins, such as Bcl-2 and Bcl-xL, providing a survival advantage to cancer cells in an oxidative stress environment [28].

Activation of HIF-1 leads to the transcription of genes, such as vascular endothelial growth factor (VEGF), which promotes the formation of new blood vessels, ensuring the supply of oxygen and nutrients to the growing tumor cells. Additionally, HIF-1 regulates genes that facilitate metabolic adaptation to hypoxia, allowing tumor cells to maintain growth under low-oxygen conditions. Finally, HIF-1 also promotes the expression of genes that favor cell survival and resistance to apoptosis, providing an adaptive advantage to cancer cells [29].

Chronic inflammation is a well-established risk factor for colorectal cancer. During inflammation, proinflammatory cytokines, such as TNF-α, IL-6, and TGF-β, promote the production of ROS and RNS through the activation of various pro-oxidant enzymes, perpetuating a state of oxidative stress and tissue damage. TNF-α can activate NADPH oxidase, increasing the production of superoxide and, subsequently, H_2_O_2_. IL-6 activates the JAK/STAT3 pathway, which is involved in regulating inflammation, immunity, and cell growth. TGF-β can also increase NOS expression, raising levels of NO and RNS. This cytokine has a dual role, acting as a tumor suppressor in early stages of cancer and as a tumor promoter in advanced stages. In the context of chronic inflammation, TGF-β can facilitate epithelial–mesenchymal transition (EMT), a crucial process in tumor invasion and metastasis [30,31,32].

## 4. Tumor Microenvironment: Immune Cells and Signaling Pathways

Inflammation is the immune system’s reaction to injury, with immune cells playing a significant role in both the acute and chronic phases of IBD. These cells are crucial for the progression from chronic inflammation to tumorigenesis.

Macrophages and lymphocytes contribute to inflammation-induced mucosal damage and promote the transition from inflammation to tumor development. The proliferation and dysregulation of lymphocytes, coupled with changes in proinflammatory cytokines and anti-inflammatory mediators, are the primary drivers of intestinal inflammation.

In CRC carcinogenesis, macrophages play a crucial but complex role. M1 macrophages have proinflammatory and antimicrobial activities, while M2 macrophages participate in waste and apoptotic cell clearance and have anti-inflammatory properties [33]. During tumorigenesis, M1 promotes tumor immunity, while M2, especially those present in tumor-associated macrophages (TAMs), favor tumorigenesis and metastasis [34] (Figure 1).

In the early stages of CAC, persistent overactivation of M1 macrophages and proinflammatory responses increase the risk of carcinogenesis. M1 macrophages can suppress M2 through oxidative stress mechanisms and competition for cytokines and chemokines.

These phenotypes are also conditioned by different environmental stimuli. If there is a deficiency of glucose and amino acids in the TME, this will cause a restriction in macrophage metabolism, leading to a decrease in the production of NADPH, ROS, and succinate, metabolites that inhibit the function of M1 macrophages. On the other hand, M2 macrophages possess a functional TCA cycle and a high expression of arginase, an enzyme that, on the one hand, reduces the availability of L-arginine for NO production, and, on the other hand, increases polyamine production. It catalyzes the reaction that leads to L-ornithine, thus inducing an anti-inflammatory environment and potentially preventing the immune response. The production of anti-inflammatory cytokines, such as IL-10, can inhibit the activity of phosphoinositide 3-kinase (PI3K), a key enzyme in the PI3K/Akt/mTOR pathway, leading to a decrease in Akt phosphorylation and activation, which in turn inhibits mTORC1 activity, a negative regulator of autophagy [35] (Figure 1).

However, in later stages, M2 inhibits the function of M1 by affecting the STAT3 and PI3K/AKT pathways, facilitating angiogenesis, tissue remodeling, and the suppression of the antitumor response, contributing to the formation of an immunosuppressive TME [36,37]. There is a dynamic balance between M1 and M2 macrophages during CRC progression.

In patients with IBD, activated CD4+ and CD8+ T cells, present in both peripheral blood and intestinal mucosa, play a key role in mediating the inflammatory response. CD8+ T cells are important in fighting tumor growth and are considered the main immune effectors against cancer cells, although their responses deteriorate during tumor progression. The infiltration and function of CD8+ T cells in the TME determine resistance to tumorigenesis [38]. In the TME characterized by hypoxia, an increase in glycolytic enzymes is observed through the transductional activation of evasion and HIF-1 genes [39], which has negative and positive effects on CD8+ T lymphocytes. Among the negative effects, metabolic dysfunction occurs because tumor cells competing for glucose in the hypoxic TME can deprive CD8+ T lymphocytes of this essential energy substrate, reducing their ability to proliferate, produce cytokines, and eliminate tumor cells. Tumor acidosis occurs due to lactic acid production from tumor glycolysis, creating an acidic environment that can suppress the function of CD8+ T lymphocytes. Both hypoxia and tumor acidosis induce the expression of PD-1 in CD8+ T lymphocytes, leading to their exhaustion and dysfunction. As a consequence of tumor glycolysis, metabolites can be generated that attract myeloid-derived suppressor cells (MDSCs) and other immune suppressors, hindering the infiltration of CD8+ T lymphocytes into the tumor.

Among the positive effects, CD8+ T lymphocyte activation is observed because hypoxia can induce surface molecule expression on tumor cells (MHC molecules, ligands for PD-1 and CTLA-4 checkpoint receptors, MICA/B, and HSP70), and some CD8+ T lymphocytes undergo metabolic adaptation using alternative metabolic pathways to adapt to hypoxia, such as fatty acid beta-oxidation, glutamate metabolism, and autophagy.

Helper T cells (Th) play a crucial role in the immune response against cancer. In the TME of colon cancer, Th1, Th2, and Th17 cells exert complex and interdependent functions that influence tumor development and progression. While Th1 and Th17 cells are involved in IBD-associated inflammation, with the secretion of cytokines, such as TNF-α, IFN-γ, and IL-6 by Th1, and IL-17, IL-22, and IL-21 by Th17, Th2 cells contribute to intestinal mucosa inflammation in ulcerative colitis (UC) by secreting IL-4 [40]. In the context of CRC, Th1 cells have been observed to have protective effects, while Th2 and Th17 cells are associated with tumor promotion and angiogenesis. Furthermore, increased TNF-α and IFN-γ in the inflamed colon contribute to CAC progression [41,42] (Figure 1).

Several signaling pathways are involved in the responses of Th1, Th2, and Th17 cells to hypoxia in the TME of colon cancer. These signaling pathways play a crucial role in regulating Th cell polarization, cytokine production, and immune function. Hypoxia-inducible factor 1 (HIF-1) is a transcription factor activated in response to hypoxia. HIF-1 regulates the expression of genes, including those involved in metabolism, cell survival, and immune signaling. In Th cells, HIF-1 can promote Th17 polarization and suppress Th1 polarization. Another pathway that can decrease Th1 polarization is the mTOR pathway, which can be inhibited by hypoxia, leading to increased Th17 polarization. Hypoxia can also activate AMPK, which can lead to increased Th1 polarization and decreased Th2 polarization.

Regulatory T cells (Tregs) play a crucial role in modulating the immune response, and, in the context of CRC, they infiltrate the TME and exert potent immunosuppressive effects that promote tumor progression. They inhibit the proliferation and cytotoxic function of effector CD8+ and CD4+ T cells by secreting immunosuppressive cytokines, such as TGF-β and IL-10. They induce myeloid suppressor cells by interacting with them, leading to a suppressive phenotype that promotes tumor angiogenesis and immune evasion.

Tregs modulate a complex network of signaling pathways that contribute to immunosuppression and tumor progression in colon cancer. Tregs activate two important pathways in effector T cells: the PI3K/Akt/mTOR pathway, leading to their dysfunction and inactivation, and the NF-κB pathway, reducing their ability to produce proinflammatory cytokines and eliminate tumor cells. Additionally, they can act on tumor cells by activating the STAT3 pathway, promoting their growth and immune evasion [43].

Although the function of Tregs has been addressed in a general way, not all Tregs are the same. Within Tregs, we can find different subpopulations with unique characteristics and functions. In fact, the increase in certain Treg subpopulations may correlate with an unfavorable prognosis in oncology [44]. Foxp3+ Tregs are the most abundant in the CRC TME and are associated with a worse prognosis [45]. During CAC progression, a transient depletion of Foxp3+ Tregs occurs, resulting in tumor growth suppression, while STAT6 signaling may facilitate CAC progression by suppressing the function of Foxp3+ Tregs. On the other hand, the expansion of RORγt+ Tregs during inflammation is associated with the activation of the Wnt-β-catenin signaling pathway, thus promoting tumorigenesis [46].

The role of the adaptive response in the TME is important, but we must consider the role played by cells of the innate immune response. In IBD, the two subpopulations of NKT cells (NKT1 and NKT2) exhibit different functions and mechanisms of action, making them key players in the dynamics of the CRC TME. Studies conducted in CRC patients have shown a decrease in type 1 NKT cells and an increase in type 2 cells in the TME and the surrounding intestine in early stages, which may persist during tumor progression. While type 2 cells may exacerbate ulcerative colitis (UC) by secreting IL-13, type 1 cells can produce inflammatory cytokines that affect the intestinal barrier [47]. However, some research suggests that type 1 NKT cells may offer protection against colitis by secreting IL-9 [48]. In summary, type 1 NKT cells have both protective and pathogenic roles in IBD, while type 2 cells tend to promote intestinal inflammation [49].

Increasing evidence highlights the duality of neutrophils in the context of IBD. In addition to their well-known role as proinflammatory agents, certain neutrophil subpopulations exhibit anti-inflammatory properties. Two subpopulations of neutrophils have been described in the TME of CRC: low-density neutrophils (LDNs) or N2, with immunosuppressive function, promote tumor growth and angiogenesis, and high-density neutrophils (HDNs) or N1, which are proinflammatory and can destroy tumor cells [50]. In the early stages of CRC, N1 subtype neutrophils are typically found, but they transform into N2 as cancer progresses. This polarization is partly induced by TGF-β, a crucial factor in reprogramming type 1 neutrophils into type 2, while the polarization from N2 to N1 is induced by the combination of GM-CSF and IFN-γ [51,52].

LDN or N2 neutrophils have the ability to regulate their migration by selectively secreting cytokines, allowing them to limit the spread of inflammation. Additionally, they actively participate in the elimination of proinflammatory cells, suppress the function of cytotoxic T cells due to their production of large amounts of arginase and dendritic cells, and significantly contribute to tissue repair and regeneration processes [53]. On the other hand, apoptotic neutrophils (N1) can regulate their migration by activating macrophages and releasing specific cytokines, facilitating their own elimination and contributing to inflammation resolution [54]. However, it is important to note that neutrophils can also have a negative impact by releasing free radicals and carcinogenic substances, such as N-nitroso compounds, which increase cancer susceptibility in IBD patients [55,56]. Particularly in the CRC TME, an increasing subtype of infiltration has been observed, CD177+ neutrophils, which promote tumor growth and suppress the activity of NK cells and cytotoxic T lymphocytes, although studies have shown that patients with a high density of CD177+ neutrophils had better overall survival and disease-free survival compared to controls [57].

However, the role of tumor-associated neutrophils has not been fully elucidated. Furthermore, both M1/M2 and N1/N2 classifications should only be used for in vitro differentiated cells, as, in vivo, both macrophages and neutrophils are highly specialized and extremely heterogeneous cells in terms of their phenotypes and functions, which are continuously shaped by the tissue microenvironment.

Innate Lymphoid Cells (ILCs) play a crucial role in intestinal inflammation and inflammatory bowel disease (IBD). They are classified into three main groups: Group 1 (NK cells and ILC1), Group 2 (ILC2), and Group 3 (ILC3) [58]. In IBD, an increase in NK cells in the intestinal mucosa is observed, which may promote inflammation and T cell differentiation. However, in animal models of colitis, NK cells can also have a protective effect by inhibiting the proinflammatory activity of neutrophils [59]. ILC1 secrete interferon gamma and are associated with Crohn’s disease and ulcerative colitis, possibly contributing to sustained inflammation and carcinogenesis [60]. ILC2 are increased in diseased tissues from IBD patients and may play a role in maintaining the intestinal mucosal barrier, although their exact role is debated. In particular, IL-13, produced by these cells, seems to facilitate the differentiation of intestinal stem cells into goblet and brush cells, which are essential for rectifying intestinal damage [61]. ILC3, especially NKp44 + ILC3, produce IL-22, which protects the integrity of the intestinal barrier but can also promote cell proliferation and colorectal cancer progression. Although these cells are important in IBD and CAC, further research is needed to fully understand their functions and implications in these diseases. A brief summary of the most relevant cells and their cytokines in the TME is provided in Table 1.

## 5. Signaling Pathways

Classic signaling pathways, such as NF-κB, PI3K/AKT, and STAT3, play a crucial role in inflammation and the development of colorectal cancer associated with inflammatory bowel disease (IBD). NF-κB, a fundamental transcription factor, regulates the production of proinflammatory cytokines and adhesion molecules, as well as the activity and development of immune cells. In patients with IBD and colorectal cancer, NF-κB activation is associated with positive regulation of PIK3R3 in intestinal epithelial cells, increasing cancer susceptibility [62]. The interaction between the intestinal microbiota and intestinal epithelial cells also activates the NF-κB pathway, promoting carcinogenesis. Additionally, NF-κB plays an important role in macrophages, where modulation of its activation can influence the development of colitis and colitis-associated cancer. These findings highlight the critical role of NF-κB in inflammation and colorectal cancer in the context of IBD [63].

PI3K, a phosphoinositide kinase, is involved in multiple cell-signaling pathways, including the activation of serine/threonine-protein kinase B (AKT), a key regulator of cellular processes, such as survival, proliferation, and metabolism [64]. PI3K/AKT activation increases the production of the proinflammatory cytokine TNF-α, exacerbating inflammation and promoting the transition from ulcerative colitis (UC) to colorectal cancer (CAC) [65]. As a consequence of the activation of this pathway, changes occur in the cells of the tumoral microenvironment related to proliferation and survival, as apoptotic pathways are inhibited in epithelial cells. AKT phosphorylates and inactivates several proapoptotic proteins, such as Bad and caspase-9, and increases the activity of antiapoptotic proteins like Bcl-2 and Bcl-xL. AKT promotes the production of vascular endothelial growth factor (VEGF), which stimulates angiogenesis, ensuring adequate blood supply to the tumor. In immune cells, the PI3K/AKT pathway can modulate the activity of various immune cells; specifically, this activation is essential for the development and suppressive function of regulatory T cells (Tregs), aiding tumors in evading immune surveillance. Furthermore, it also influences the polarization of macrophages towards an M2 phenotype, associated with anti-inflammatory responses and tissue repair, thus supporting tumor growth and metastasis [66,67,68].

β-catenin signaling induced by PI3K/AKT also plays a crucial role in the activation of progenitor cells during the progression from UC to CAC. Additionally, the inhibition of these pathways can reduce intestinal inflammation and tumorigenesis [69].

STAT3, another key transcription factor, is activated in response to inflammatory cytokines, such as IL-6 and IL-22, promoting cell viability and CAC progression [70]. In addition to activation in tumor cells, STAT3 signaling is essential for the differentiation of Th17 cells, inhibition of dendritic cell maturation, and maintenance of the immunosuppressive function of Foxp3+ Treg cells. Constitutive activation of STAT3 in various immune cells infiltrating tumors, such as dendritic cells and macrophages, will lead to sustained secretion of proinflammatory cytokines and shift the local microenvironment towards an immunosuppressive direction [71].

STAT3 activation by most signaling molecules is primarily transduced through activation of STAT3 by IL-6. The Notch pathway acts as an effector of IL-6/STAT3 signaling, regulating cellular self-renewal, differentiation, and tumorigenic properties [72]. Recent studies have demonstrated that STAT3 signaling axis activation through IL-11 production by cancer-associated fibroblasts and myeloid cells is more potent for gastrointestinal tumor progression than IL-6 [73]. IL-11+ fibroblasts promote tumor progression by secreting IL-11 to activate colon tumor epithelial cells and colon fibroblasts [74]. STAT activation by IL-11 in cancer-associated fibroblasts often indicates a poor prognosis [75].

The Wnt/β-catenin signaling pathway regulates the inflammatory cascade response and oxidative stress, favoring tumor development and progression. Hyperactivation of the Wnt/β-catenin signaling pathway has been associated with the development of CRC. There is increasing evidence that mutations in critical regulators of the Wnt/β-catenin signaling pathway are associated with the majority of CRC cases [76]. β-catenin is a critical component of the Wnt signaling pathway. As the central component of the Wnt/β-catenin signaling pathway, β-catenin plays a crucial role in Wnt signal transduction to activate gene transcription leading to the pathological phenotype of CRC, such as proliferation and metastasis [77]. WNT signaling stabilizes β-catenin, allowing it to translocate into the nucleus and activate the transcription of genes involved in cell proliferation and differentiation, resulting in uncontrolled cell division and tumor growth. It is important to note that WNT/β-catenin signaling is crucial for maintaining the self-renewal capacity of cancer stem cells and facilitating the epithelial–mesenchymal transition (EMT), thereby increasing the invasive and metastatic potential of tumor cells. Additionally, it also affects dendritic cells (DCs) by leading them to a tolerogenic state, affecting their ability to activate T cells, thus promoting immune evasion by the tumor. Finally, the influence of this pathway on the tumor microenvironment should be noted, as it modulates the activity of fibroblasts and other stromal cells, creating an environment that supports tumor growth and protects the tumor from immune attacks [78,79,80].

Although there are currently conflicting reports on the role of β-catenin in IBD-associated CRC, considering the role of the intestinal microbiota, it has been found that the effect of specific gut microbiota on IBD-associated CRC arises, at least in part, from the activation of β-catenin signaling.

In the context of chronic intestinal inflammation, the IL-23/Th-17 axis emerges as a crucial player in tumor development promotion. The disruption of the epithelial barrier, characteristic of this inflammatory condition, triggers dendritic cell activation by microbial products. These activated dendritic cells, in turn, secrete IL-23, a cytokine that induces differentiation and activation of Th17 cells in the lamina propria. The IL-6/STAT3 pathway plays a fundamental role in this IL-23-mediated differentiation process [81]. Th17 cells activated by IL-23 release proinflammatory cytokines, such as IL-17A and IL-17F, which, through activation of the STAT3/NF-κB pathway, stimulate the production of additional proinflammatory cytokines, including IL-6 and TNF-α [82,83].

This cascade of proinflammatory cytokines amplifies the preexisting inflammatory response, creating a procarcinogenic microenvironment that promotes tumor growth and progression.

TNF Axis: Sustained low-level production of TNF-α can induce a tumorigenic phenotype [84]. This tumorigenic effect of TNF-α is based on the generation of ROS and RNS, which cause DNA damage and promote tumor formation [85]. In the presence of TNF-α and IL-1, the PI3K/AKT signaling pathway activates NF-κB by phosphorylating IKK [86]. On one hand, the interaction of the TNF-α/TNF receptor axis leads to massive local infiltration of macrophages and neutrophils through the release of chemokines. This, in turn, activates inflammatory pathways, such as NF-κB, and regulates the production of inducible nitric oxide synthase (iNOS). On the other hand, TNF-α can also promote tumor angiogenesis by inducing infiltration of macrophages and neutrophils expressing COX-2.

These signaling pathways are fundamental in both inflammation and cancer, suggesting that therapeutic agents targeting these pathways could be effective in the treatment of CCR. However, further studies are needed, especially in patients with CCR at various stages, to fully understand the molecular mechanisms and develop specific therapeutic approaches. Table 2 summarizes the main signaling pathways involved in colorectal cancer carcinogenesis.

## 6. Microbiota and CRC

In recent years, numerous external factors have been delineated that can either elevate or diminish the likelihood of developing CRC. Among these is the microbiome. Comprising the entirety of microbial genes in the human intestine, it is an essential component of the intestinal microbiota. This denotes the assemblage of microorganisms that inhabit this habitat [87]

The microbiome of the human and animal intestines comprises a diverse array of microorganisms, spanning bacteria, archaea, fungi, viruses, and multicellular parasites. [88]. The predominant constituents of intestinal bacteria primarily consist of four bacterial phyla: Firmicutes, Bacteroidetes, Proteobacteria, and Actinobacteria [89]. The significance of the human intestinal microbiome in the context of IBD has been progressively acknowledged. It is principally distinguished by an escalation in pathogenic bacterial strains juxtaposed with a reduction in advantageous bacterial communities [90,91].

The well-being of the microbiome plays a pivotal role in immune system maturation and the modulation of immune reactions. Conversely, dysbiosis is linked to a spectrum of ailments, including IBD and CRC. Therefore, the bacterial composition in healthy tissue differs significantly from that found in patients with CRC. Through the examination of fecal specimens sourced from individuals exhibiting good health, those diagnosed with IBD, and patients afflicted with CRC, particular bacterial species have been discerned, demonstrating an association with the pathogenesis of these conditions [92,93,94].

Several mechanisms could elucidate the bacterial involvement in CRC development (see Figure 2). Primarily, bacteria and their metabolites may exhibit genotoxicity, directly transforming intestinal epithelial cells (IECs). An exemplary instance involves the existence of *Escherichia coli* harboring the genomic island polyketide synthase (pks+ *E. coli*), responsible for the synthesis of colibactin and the production of potentially mutagenic DNA adducts [95]. Preclinical models of CRC and human tumor tissues exhibit an abundance of pks+ *E. coli* [96]. Furthermore, these microorganisms elicit a distinct mutational imprint in ex vivo cultured non-transformed colon organoids, a pattern also discerned among CRC patients [97]. Moreover, the microbial synthesis of gallic acid might exert a localized impact on the functionality of mutant tumor suppressor proteins, such as p53, within the intestinal epithelium, thereby implying microbiome-mediated control over the oncogenic propensity of specific mutations [98].

In second place, the compromised integrity of the superficial barrier function within CRC tumors may initiate inflammation induced by commensal bacteria, thereby fostering tumorigenesis. For instance, the breakdown of tight junctions among colon tumor cells permits the ingress of degradation by products from commensal bacteria, such as lipopolysaccharide, into the tumor stroma, consequently prompting the recruitment of myeloid cells in the TME. Furthermore, commensal bacteria can infiltrate tumor tissue themselves, stimulating infiltrating myeloid cells to generate inflammatory cytokines and thereby exacerbating CRC tumorigenesis.

Thirdly, it is noteworthy that pathogenic bacteria have the capacity to instigate inflammation within the colon, consequently fostering tumorigenesis. For instance, *Fusobacterium nucleatum* is a Gram-positive anaerobic bacterium, commensal and typically present in the oral cavity, but it is not commonly found in other parts of the body under normal conditions. Nevertheless, the human colorectal cancer microenvironment manifests a discernible augmentation of *F. nucleatum* [99,100], concomitant with a prognostic decrement [101]. *F. nucleatum* facilitates tumorigenesis through its interactions with both tumor cells and immune cells present within the tumor microenvironment. As an illustration, *F. nucleatum* prompts β-catenin signaling within colonic epithelial cells, thereby amplifying the expression of oncogenes, including cyclin D1 and MYC, alongside instigating proinflammatory signals, such as TNF and IL-17 [102]. In the ApcMin/+ mouse model, the presence of *F. nucleatum* is associated with an elevated occurrence of CRC tumors. Moreover, it leads to an enhanced recruitment of MDSCs, TANs, TAMs, and immature dendritic cells possessing immunosuppressive capabilities, all of which collectively drive tumor progression [103]. Moreover, *F. nucleatum* engages with the inhibitory immune receptor TIGIT, thereby dampening T cell activation and impairing natural killer (NK) cell-mediated clearance of colon tumor cells [104]. Additionally, the presence of *F. nucleatum* triggers the expression of specific chemokines, such as CXCL8 and CXCL1, consequently amplifying the migratory capability of tumor cells and reinforcing metastatic potential in both infected and uninfected cells [105]. In contrast to *F. nucleatum*, which can drive tumorigenesis independently of colitis, enterotoxigenic Bacteroides fragilis (ETBF) initially provokes colitis and subsequently facilitates tumorigenesis [106]. ETBF also orchestrates the recruitment of other bacterial species to form biofilms that coat colorectal adenomas and tumors [107], thereby initiating a proinflammatory cascade within colonic epithelial cells, thus facilitating the progression of CRC [108]. Furthermore, ETBF demonstrates the capability to enlist myeloid-derived suppressor cells (MDSCs) to the TME, thereby exacerbating the suppression of antitumor immune responses in CRC [109].

In addition, it is essential to mention the beneficial bacteria of the intestinal microbiome, which also play a crucial role. Reference should be made to *Akkermansia muciniphila* and *Faecalibacterium prausnitzii*, belonging to the bacterial phylum Firmicutes, whose presence may influence the local immune response in the TME of CRC [110].

The presence of these bacteria is associated with increased infiltration of antitumor immune cells, such as cytotoxic T lymphocytes and dendritic cells, into the tumor tissue. This suggests a potential role of these bacteria in modulating the antitumor immune response in the CRC TME [110].

*A. muciniphila* is a commensal bacterium found in the intestinal mucosa, and it has been associated with improved metabolic health, including blood glucose regulation and obesity prevention. It can also strengthen the integrity of the intestinal barrier and modulate the immune response, maintaining immune system homeostasis, suggesting a protective role in intestinal health. Recent studies have suggested that the presence of *A. muciniphila* may be associated with a better antitumor immune response in the CRC TME, which could have implications for disease progression and prognosis [111,112].

Although the exact mechanisms are not yet fully elucidated, some cellular and molecular pathways that could be involved in these processes have been identified. 

At the molecular level, it has been demonstrated that *A. muciniphila* exerts beneficial effects through various pathways. For example, this bacterium can degrade mucin, which is an essential component of the intestinal mucus, using specialized enzymes, such as sulfatases and glycosidases [113]. This ability to degrade mucin allows it to colonize the mucosal layer of the intestine and promote intestinal barrier integrity [114].

Furthermore, recent studies have revealed that *A. muciniphila* may modulate the immune response by inducing anti-inflammatory cytokines. It has been observed that the presence of this bacterium is associated with an increase in the production of anti-inflammatory factors, such as transforming growth factor beta (TGF-β) and interleukin-10 (IL-10), which can modulate immune cell activity and promote an anti-inflammatory environment in the intestine [115,116].

All of the aforementioned information is correlated with evidence that its presence is diminished in patients with CRC compared to healthy individuals [111].

On the other hand, *F. prausnitzii* is a butyrate-producing bacterium, a short-chain fatty acid with anti-inflammatory properties that constitutes an important energy substrate for colon epithelial cells and helps maintain intestinal barrier integrity. The decrease in *F. prausnitzii* abundance has been observed to be associated with chronic inflammation and various intestinal diseases, including CRC. Recent studies have suggested that the presence of *F. prausnitzii* in CRC tumor tissue could be associated with reduced inflammation, increased infiltration of antitumor immune cells, and a better response to treatment, suggesting a possible protective role in cancer progression [111,117,118,119].

At the molecular level, it has been proposed that F. prausnitzii may modulate the immune response and inflammation in the CRC TME through various pathways. For instance, it has been suggested that the production of butyric acid by F. prausnitzii can inhibit the production of proinflammatory cytokines and promote the differentiation of regulatory T cells in the intestine, thus contributing to an anti-inflammatory environment and promoting the antitumor immune response [120].

In summary, the role of beneficial bacteria, such as *A. muciniphila* and *F. prausnitzii*, in the context of CRC is of utmost importance. These bacteria not only contribute to maintaining intestinal health and immune system homeostasis but also play a crucial role in modulating the antitumor immune response and inflammation in the CRC TME.

Certain bacteria, in collaboration with particular constituents of the diet, play a crucial role in modulating the immune system, either by promoting or mitigating intestinal inflammation and the onset of tumors via the generation of metabolites or co-metabolites [121]. The gastrointestinal microbiome generates a wide variety of metabolic compounds through the metabolic transformation of externally sourced dietary constituents, endogenous microbial compounds residing within the intestinal lumen, or those derived from the host.

Upon traversing the mucosal barrier, these compounds have the capability to directly interact with intestinal epithelial cells (IECs) or exert an impact on the immune reactions within the intestinal stromal tissue. Prominent instances of these metabolites encompass short-chain fatty acids (SCFAs) like butyric acid, acetic acid, and propionic acid. These compounds stand as the primary metabolic by-products resulting from the bacterial fermentation process of incompletely digested complex carbohydrates. These SCFAs, beyond serving as a vital energy source for intestinal microorganisms and IECs, play diverse regulatory roles in host physiology and immunity, and they are commonly characterized by their antitumorigenic effects. Studies have indicated that SCFAs exert inhibitory effects on histone deacetylases (HDACs) within neutrophils, resulting in a reduction in the synthesis of TNF and NO, along with the suppression of nuclear factor kappa B (NF-κB) signaling. Comparable anti-inflammatory outcomes may be discerned within macrophages, wherein short-chain fatty acid (SCFA) derivatives mitigate the synthesis of proinflammatory agents, such as IL-6, IL-12, and NO, within the gastrointestinal tract [122]. Moreover, AGCCs might obstruct the differentiation process of dendritic cells originating from bone marrow and augment the suppressive potential of regulatory T cells expressing the FOXP3 marker by inhibiting enzymes responsible for histone deacetylation [123]. Contrastingly, *E. coli* amplifies the functionality of HDAC3 in intestinal epithelial cells via phytate metabolism and inositol-1,4,5-trisphosphate production, thus facilitating the restoration post intestinal injury through the modulation of epithelial cell growth [124]. The intestinal microbiota also holds considerable sway over the CRC response to therapy, notably in the realm of immunotherapy [125]. Following traditional chemotherapy, which induces programmed cell death of epithelial cells in the ileum, microorganisms inhabiting the ileum, such as *Bacteroides fragilis* and *Erysipelotrichaceae*, communicate with migratory dendritic cells to stimulate follicular helper T cells residing in tertiary lymphoid structures linked to colorectal tumors. In collaboration with B cells, they enhance the antitumorigenic outcomes of chemotherapy and immunotherapy [126].

## 7. Discussion

The transformation of inflammation induced by IBD into carcinogenesis is a complex process involving intricate interaction among immune cells, their metabolic products, signaling pathways, and the intestinal microbiome. In this context, the functional duality of immune cells in the TME emerges as a crucial factor modulating cancer progression and response to therapy.

Immune cells, such as macrophages and T cells, exhibit polarization into different subtypes depending on the cytokines and signaling pathways activated or inhibited within the TME. M1 macrophages, characterized by their production of proinflammatory cytokines and ROS, act as allies of the immune system, eliminating tumor cells. However, in the CRC TME, these macrophages can polarize towards an M2 phenotype, characterized by their production of anti-inflammatory cytokines, promoting tumor immune evasion and angiogenesis. T cells also exhibit functional duality. Cytotoxic effector CD8+ T cells are essential for direct elimination of tumor cells, while Foxp3+ Treg cells act as negative regulators of the immune response, suppressing effector T cell activity and promoting tumor tolerance. The balance between these polarized T cells determines the efficacy of the antitumor immune response.

Signaling pathways, such as the NF-κB pathway and the JAK/STAT pathway, play a fundamental role in immune cell polarization and regulation of the antitumor immune response in CRC. NF-κB pathway activation in macrophages favors M1 polarization, while its inhibition favors M2 polarization. The JAK/STAT pathway, on the other hand, regulates the differentiation and function of T cells, including the activity of Foxp3+ Treg cells.

The role of inflammation in predicting cancer outcomes and the potential clinical application of anti-inflammatory treatments in colorectal cancer are significant areas of research. Chronic inflammation has been associated with the development and progression of various types of cancer, including colorectal cancer. This chronic inflammatory process generates a tumoral microenvironment that promotes cell proliferation, local invasion, and distant metastasis. Various molecular mechanisms participate in this process, including the activation of the nuclear factor kappa-B (NF-κB) signaling pathway. Throughout this review, the crucial role of NF-κB activation has been demonstrated and how its dysregulation is implicated in colorectal cancer. Therefore, targeting NF-κB activity with anti-inflammatory agents could be a promising approach in cancer therapy. Nonsteroidal anti-inflammatory drugs (NSAIDs), such as aspirin and celecoxib, have shown antitumor properties in preclinical and clinical studies. These drugs can inhibit NF-κB pathway activity, resulting in reduced cell proliferation, induction of apoptosis, and suppression of tumoral angiogenesis. Among them, sulindac and aspirin are NSAIDs that have been investigated for their potential application in the treatment of colorectal cancer. Sulindac has been shown to be effective in reducing the risk of colorectal adenomatous polyps, precursors of colorectal cancer. Aspirin has also shown a preventive effect on colorectal cancer, especially in patients with a family history of the disease [127,128,129,130,131].

Research into the role of the microbiome and microbiota in CRC has emerged as a crucial field, providing deeper insights into the complex interaction between intestinal bacteria and tumor development [87]. It has been evidenced that the composition of the intestinal microbiome can significantly vary between healthy individuals and CRC patients, suggesting a potential role in disease pathogenesis. Dysbiosis, characterized by an imbalance in bacterial composition, has been associated with various intestinal diseases, including CRC, highlighting the importance of understanding the implications of these bacteria in health and disease.

Among the bacteria studied in the context of CRC, both those considered pathogenic and those considered beneficial for intestinal health have been scrutinized. For instance, bacteria, such as *Escherichia coli*, *Fusobacterium nucleatum*, and *Bacteroides fragilis*, have been associated with CRC and can influence the TME and the immune response [107,132,133]. The presence of these pathogenic bacteria can trigger inflammatory processes and promote carcinogenesis, suggesting that selective elimination of these bacteria could be a potential therapeutic strategy.

However, it is imperative to consider the potential systemic side effects of eliminating these pathogenic bacteria. Given that the intestinal microbiome plays a fundamental role in homeostasis and various physiological functions, selective elimination of certain bacterial species could disrupt microbial balance and trigger unwanted side effects, such as metabolic dysfunction or inappropriate immune responses [134].

On the other hand, there are bacteria considered beneficial for intestinal health, such as *A. muciniphila* and *F. prausnitzii*, which have demonstrated protective effects against CRC development [135]. These bacteria not only contribute to maintaining intestinal health and immune system homeostasis but also play a crucial role in modulating the antitumor immune response and inflammation in the CRC tumor microenvironment.

Supplementing the intestine with these beneficial bacteria could represent a promising therapeutic strategy to prevent CRC development or improve clinical outcomes in patients with this disease. However, it is essential to consider the potential negative side effects of this intervention, as altering the intestinal microbiome could affect other physiological functions and trigger unexpected host responses.

To fully understand the underlying mechanisms and clinical potential of the intestinal microbiome in CRC, a multidisciplinary approach is required, integrating advanced research tools, such as network biology studies and GO and KEGG analyses. These tools allow for exploration of how signaling pathways and biological processes are interconnected and how inhibition or supplementation of certain bacteria could alter physiology and trigger cascade effects through multiple cellular pathways. This is substantiated by various studies, among which we can highlight, on the one hand, a multi-omics study on CRC patients published in the *Journal of Translational Medicine*. The study analyzed metagenomic sequencing of fecal samples and exome and transcriptome sequencing of mucosal tissue in CRC patients. Using HUManN2 to estimate the relative abundance of KEGG ontology categories, changes in bacterial metabolic pathways associated with CRC were identified, such as the “one carbon pool by folate” pathway, which was significantly elevated in CRC patients. This pathway is essential for the cellular metabolic process supporting tumorigenesis [136].

On the other hand, another study published in *Microbiome* conducted 16S rRNA gene sequencing and transcriptomic analyses to investigate the biological functions of the gut microbiome in patients with lymphovascular invasion (LVI) in CRC. This study included functional enrichment analysis of transcriptomes related to LVI using tools, such as GO and KEGG, to identify non-invasive biomarkers and to construct diagnostic models for assessing LVI conditions in CRC [137].

Finally, another study published in *Microbiome* performed a meta-analysis of the CRC metagenome, identifying altered bacteria in different populations and universal bacterial markers. The study utilized the SparCC algorithm to investigate correlations between bacterial species enriched and depleted in CRC, demonstrating that CRC-related bacteria formed mutually exclusive networks. The analysis of bacterial correlation networks elucidated how these interactions might influence CRC progression [138].

These studies, among many others, exemplify how the integration of GO and KEGG analyses with systems biology approaches can provide a detailed understanding of the molecular mechanisms of the gut microbiome in CRC and its potential clinical impact.

In conclusion, while the intestinal microbiome plays a significant role in CRC carcinogenesis, with the presence of both pathogenic and beneficial bacteria influencing the tumor microenvironment and immune response, further research is needed to fully understand their underlying mechanisms and clinical potential. Continued study of the intestinal microbiome in the context of CRC could lead to new opportunities for the development of personalized therapeutic approaches and improved clinical outcomes in patients with this disease.

## 8. Conclusions

Understanding the functional duality of immune cells in the TME and the effects of toxic products, reactive oxygen and nitrogen intermediates, and cytokines is essential for the development of new therapeutic strategies that modulate the immune response and improve cancer patient prognosis. These strategies could include modulation of ROS and RNS production, blockade of anti-inflammatory cytokines, or stimulation of effector T immune cells.

Immunometabolism, a burgeoning discipline, investigates the interaction between immunity and metabolism. During the chronic inflammation phase, administration of anti-inflammatory and immunomodulatory drugs may help prevent cancer formation. However, at the tumor stage, it is crucial to enhance antitumor immunity to successfully suppress tumor growth. The TME can be improved by optimizing immune cell recruitment through metabolic modulation. Metabolites produced by tumor cells and immune cells impact immune cell activation and phenotypic changes, as well as the regulation of immune checkpoints.

Recent studies have underscored the critical importance of the microbiome in colorectal cancer (CRC), highlighting its potential as a target for therapeutic interventions. Specific bacteria, such as *Fusobacterium nucleatum*, *Bacteroides fragilis*, and *Escherichia coli*, have been identified as being associated with inflammation and CRC progression, making them key therapeutic targets. Interventions including fecal microbiota transplantation (FMT), dietary modifications, and synthetic biology are showing promise in modulating the microbiome, reducing inflammation, and enhancing the efficacy of conventional therapies, such as chemotherapy and immunotherapy. These approaches not only improve treatment response but also reduce side effects, offering new avenues for more personalized and effective management of CRC.

## Figures and Tables

**Figure 1 ijms-25-06188-f001:**
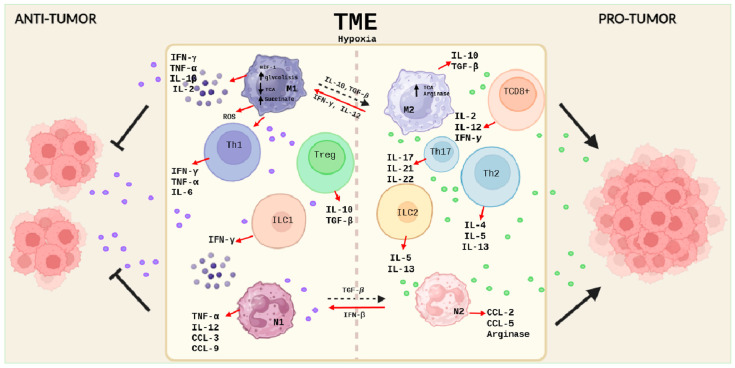
Schematic description of the cellular components of the immune system in the TME of colorectal cancer. On the (**left**), the cells of the immune system and their secreted products involved in the antitumor response are represented, and, on the (**right**), the cells and molecules involved in a pro-tumor response are represented. Created with BioRender.com.

**Figure 2 ijms-25-06188-f002:**
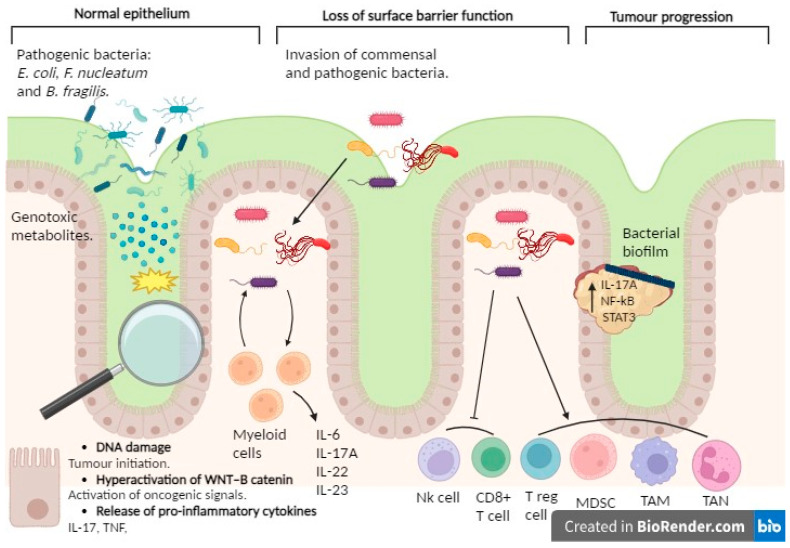
Involvement of the microbiome in CRC. The involvement of the microbiome in colorectal cancer becomes apparent through the capacity of specific pathogenic bacteria, such as *Escherichia coli*, *Fusobacterium nucleatum*, and *Bacteroides fragilis*, alongside their metabolites, to incite tumorigenesis. This initiation may ensue through direct mutagenic impacts on intestinal epithelial cells and the triggering of intracellular signaling pathways associated with oncogenesis, notably the hyperactivation of the WNT–β-catenin pathway. Furthermore, bacterial metabolites may provoke inflammatory responses by releasing factors like TNF and IL-17, thereby exacerbating tumor formation. Diminished intestinal barrier function may facilitate bacterial invasion from the intestinal lumen, prompting an inflammatory response that fosters tumor growth. These bacteria can form biofilms around the tumor, activating inflammatory signaling pathways (involving factors like NF-κB, STAT3, and the IL-17A receptor) that promote CRC development. Moreover, these bacterial entities exhibit the ability to quell immune responses within the TME through the recruitment of immunosuppressive cell populations, including myeloid-derived suppressor cells (MDSCs) and regulatory T cells (Tregs). Additionally, they orchestrate the polarization of immune cells towards phenotypes conducive to tumor expansion and immune dampening, typified by tumor-associated macrophages (TAMs) and tumor-associated neutrophils (TANs). Created with BioRender.com.

**Table 1 ijms-25-06188-t001:** Inflammatory and tumor-associated cells and their cytokines in the colorectal cancer microenvironment.

Pro-Tumor Cells	Associated Cytokines	Antitumor Cells	Associated Cytokines
Macrophages M2	IL-10, TGF-β	Macrophages M1	INF- γ, TNF-α, IL-1β, IL-2
T helper 2 cells (Th2)	IL-4, IL-5, IL-13	T helper 1 cells (Th1)	IFN-γ, TNF-α, IL-6
T helper 17 cells (Th17)	IL-17, IL-21, IL-22	T regulatory cells (Treg)	IL-10, TGF-β
Innate Lymphoid Cells 2 (ILC2)	IL-15, IL-13	Innate Lymphoid Cells (ILC1)	INF-γ
Neutrophils N2	CCL-2, CCL-5 Arginase	Neutrophils N1	TNF-α, IL-12, CCL-3, CCL-9
T Cytotoxic lymphocytes (TCD8+)	IL-2, IL-12, IFN-γ		

**Table 2 ijms-25-06188-t002:** Signaling pathways involved in colorectal cancer carcinogenesis.

Signaling Pathway	Description	Associated Cytokines and Factors	Effects on Carcinogenesis
NF-κB	The NF-κB pathway is a key regulator of the immune response and inflammation. It is activated in response to cytokines, oxidative stress, and microbial products.	TNF-α IL-1β IL-6	Promotes cell survival, proliferation, angiogenesis, and resistance to apoptosis. Contributes to an inflammatory environment favoring tumor growth.
PI3K/AKT	The PI3K/AKT pathway is involved in the regulation of cellular growth, proliferation, survival, and metabolism. It is activated by tyrosine kinase receptors and cytokines.	IGF EGF IL-6	Promotes cell survival and proliferation, inhibits apoptosis, and contributes to treatment resistance. Implicated in angiogenesis and cellular metabolism.
STAT3	The STAT3 pathway is activated by cytokines and growth factors, such as IL-6 and EGF. It is a transcription factor that regulates the expression of genes involved in cell proliferation and survival.	IL-6 IL-22	Promotes cell proliferation, angiogenesis, resistance to apoptosis, and evasion of the immune system. Contributes to chronic inflammation and tumor development.
Wnt/β-catenin	The Wnt/β-catenin pathway regulates cellular proliferation and differentiation during development. In cancer, abnormal activation of this pathway leads to the accumulation of β-catenin in the nucleus, where it activates the transcription of genes promoting tumor growth.	Wnt ligands β-catenin	Promotes cell proliferation, invasion, and migration, and is associated with tumor progression and resistance to apoptosis.

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
