# Peer review of "The Crucial Role of Inflammation and the Immune System in Colorectal Cancer Carcinogenesis: A Comprehensive Perspective"

_ijms, 2024, doi:10.3390/ijms25116188_

Round 1
Reviewer 1 Report
Comments and Suggestions for Authors
Summary:
The goal of this review manuscript is to summarize the current understanding of the relationship between chronic inflammation and the carcinogenesis of colorectal cancer, with a specific emphasis on how pro-inflammatory cytokines affect the tumor microenvironment and genetic/epigenetic factors predisposing to cancer.
Sections 1 and 2 start with a brief overview of the incidence of colorectal cancer (CRC) before briefly discussing chronic inflammatory diseases of the bowel. Some specific inflammatory factors, including TNF-a, IL-6, and IL-1B, are mentioned along with their role in altering the tumor microenvironment and tumor growth. Section 3 goes into more detail on these processes, in particular the role of ROS and RNS.
Section 4 discusses the role of inflammatory cells in controlling the tumor microenvironment, in particular the activation of lymphocytes and certain macrophage pathways. Substantial time is spent discussing the pro- or anti-inflammatory roles of M1 and M2 macrophages, CD4+ and CD8+ T cells, helper T cells, Tregs, and other inflammatory cells. Section 5 details the important signaling pathways of NFkB, PI3K/AKT, and STAT3 as well as their role in chronic inflammation and tumorigenesis.
Section 6 discusses the gut microbiome and its associations with chronic inflammation of the bowel as well as gastrointestinal tumors. The general mechanisms of how bacteria in the gut can affect the mucosa and elicit inflammation are discussed, as well as several specific bacteria that have been associated with intestinal health, regulation of the local immune response, and cell proliferation.
The Discussion and Conclusions sections provide another quick summary of the main points discussed in the text of the paper, and the authors discuss several important areas for future research into mechanisms of inflammation and carcinogenesis as well as research areas for therapeutic interventions.
General Comment:
Overall, I believe that this is a great review of several very important topics regarding inflammation and cancer of the colon. The organization of topics is reasonable and allows the reader to easily follow the authors’ points. Overall, the content of the review and the level of detail are very good, and I believe this paper is acceptable for publication with minimal changes.
Section 4 in particular is an excellent overview of many different inflammatory pathways. As a review article, having an additional table in this section to summarize the different cell types into pro-inflammatory vs anti-inflammatory and/or pro-tumor vs anti-tumor would be greatly beneficial to the reader. Similarly, Section 5 is also an excellent overview of the important signaling pathways of cellular proliferation and inflammation that are implicated in carcinogenesis; another table in this section would be greatly appreciated as well.
One critique I have is that the frequency of citations is a little inconsistent, and there are a few specific points that are listed without adequate citations (detailed later in this review).
The figures are well-made, and their inclusion strengthens the paper, although they are small and difficult to read in the review version of the manuscript.
There seems to be an error in numbering the sections. The sections in the paper are numbered 1, 2 3, 4, 5, 6, then an additional section 3 (Discussion) and then 5 (Conclusions). Are sections 3-6 intended to be printed as subsections of “2. Colorectal Cancer and Intestinal Disease”?
The paper is well-written with excellent English at the level of a native speaker. As a minor point, the paper frequently uses both “signaling” (American English) and “signalling” (British English), and the authors’ should be consistent in which form they use.
Specific questions and recommendations by section:
Section 1:
Line 39-40: Although this is broadly true, having familial relatives with a history of CRC is still one of the biggest risk factors for developing CRC (with some studies showing it is significantly more impactful than any one specific environmental factor), and this sentence as written seems to discount this fact. A more accurate way to phrase this sentence would be to say “rather than any specific heritable genes” instead of “rather than hereditary genetic changes.” Please include a citation here as well.
Section 2:
Although the information in Section 2 is correct, citations should be added after a few more points (lines 61-62 for example).
Colorectal cancer is a common neoplasm, but is the incidence of colorectal cancer increasing? What is the incidence of chronic inflammatory diseases such as Ulcerative Colitis, IBD, or Crohn’s Disease? Do general trends in the incidence of such diseases correlate with the incidence of CRC? Adding this information to sections 1 or 2 would be of interest to the reader and strengthen the paper.
Section 3:
Line 103: Reactive Oxygen and Nitrogen Species should be explicitly mentioned before use of RONS as an acronym (RONS is used in line 103 before it is defined in lines 104-105).
Section 4:
Line 130: “metástasis” should be “metastasis”
Figure 1: Please specify what the dotted lines mean.
Line 163: The italics are not necessary for “MYC and HIF-1 genes.”
Line 209: “evasión” should be “evasion”
Line 267: “carcinogénesis” should be “carcinogenesis”
Section 5:
Line 324: Please add a citation here (such as #89 based on discussions in section 6).
Author Response
Reviewer 1
Dear Reviewer, first of all, thank you for taking the time to review our manuscript. We tried to address all your suggestions. Please find the answers to your comments in the following paragraphs:
- 2 tables have been included in section 4 and 5, as suggested by reviewer 1.
Table 1. Inflammatory and tumor-associated cells and their cytokines in the colorectal cancer microenvironment
|
Pro-tumor Cells |
Associated Cytokines |
Anti-tumor Cells |
Associated Cytokines |
|
Macrophages M2 |
IL-10, TGF-β |
Macrophages M1 |
INF- γ, TNF-α, IL-1β, IL-2 |
|
T helper 2 cells (Th2) |
IL-4, IL-5, IL-13 |
T helper 1 cells (Th1) |
IFN-γ, TNF-α, IL-6 |
|
T helper 17 cells (Th17) |
IL-17, IL-21, IL-22 |
T regulatory cells (Treg) |
IL-10, TGF-β |
|
Innate Lymphoid Cells 2 (ILC2) |
IL-15, IL-13 |
Innate Lymphoid Cells (ILC1) |
INF- γ |
|
Neutrophils N2 |
CCL-2, CCL-5 Arginase |
Neutrophils N1 |
TNF-α, IL-12, CCL-3, CCL-9 |
|
T Cytotoxic lymphocytes (TCD8+) |
IL-2, IL-12, IFN-γ |
|
|
Table 2. Signalling Pathways Involved in Colorectal Cancer Carcinogenesis
|
Signalling Pathway |
Description |
Associated Cytokines and Factors |
Effects on Carcinogenesis |
|
NF-κB |
The NF-κB pathway is a key regulator of the immune response and inflammation.
It is activated in response to cytokines, oxidative stress, and microbial products. |
TNF-α IL-1β IL-6 |
Promotes cell survival, proliferation, angiogenesis, and resistance to apoptosis.
Contributes to an inflammatory environment favouring tumour growth. |
|
PI3K/AKT |
The PI3K/AKT pathway is involved in the regulation of cellular growth, proliferation, survival, and metabolism.
It is activated by tyrosine kinase receptors and cytokines. |
IGF EGF IL-6 |
Promotes cell survival, proliferation, inhibits apoptosis, and contributes to treatment resistance.
Implicated in angiogenesis and cellular metabolism. |
|
STAT3 |
The STAT3 pathway is activated by cytokines and growth factors such as IL-6 and EGF.
It is a transcription factor that regulates the expression of genes involved in cell proliferation and survival. |
IL-6 IL-22 |
Promotes cell proliferation, angiogenesis, resistance to apoptosis, and evasion of the immune system.
Contributes to chronic inflammation and tumour development. |
|
Wnt/β-catenin |
The Wnt/β-catenin pathway regulates cellular proliferation and differentiation during development. In cancer, abnormal activation of this pathway leads to the accumulation of β-catenin in the nucleus, where it activates the transcription of genes promoting tumour growth. |
Wnt ligands β-catenin |
Promotes cell proliferation, invasion, migration, and is associated with tumour progression and resistance to apoptosis. |
- The figures have been enlarged to improve readability and comprehension.
- The sections have been renumbered in a logical order.
- The term ‘signalling’ has been corrected throughout the manuscript by adopting the British English version, (marked in the text in red).
Recommendations by section:
Section 1.
Incidence data on inflammatory bowel diseases have been included as suggested by reviewer 1 (lines 38 to 50).
As suggested by reviewer 1, the phrase ‘rather than hereditary genetic changes’ has been replaced by the phrase originally in the manuscript ‘rather than any specific heritable genes’. Citation included.
Section 2.
One citation has been included in section 2: Grivennikov SI, Greten FR, Karin M. Immunity, inflammation, and cancer. Cell. 2010 Mar 19;140(6):883-99. doi: 10.1016/j.cell.2010.01.025.
Section 3.
RONS has been defined in line 103.
Section 4.
The typographical errors specified in lines 130, 163, 209 and 267 and figure 1 have been corrected.
Section 5.
One citation has been included in line 324: Wang Y, van Boxel-Dezaire AH, Cheon H, Yang J, Stark GR. STAT3 activation in response to IL-6 is prolonged by the binding of IL-6 receptor to EGF receptor. Proc Natl Acad Sci U S A. 2013 Oct 15;110(42):16975-80. doi: 10.1073/pnas.1315862110. Epub 2013 Sep 30.

Reviewer 2 Report
Comments and Suggestions for Authors
This paper provides a comprehensive perspective on the role of inflammation and the immune system in CRC carcinogenesis. By addressing the following points, the paper will be more comprehensive and impactful.
Section 3:
Currently, this section broadly discusses the role of ROS and RNS in carcinogenesis. The authors should focus specifically on the role of ROS and RNS in colorectal cancer carcinogenesis. They should discuss how ROS and RNS contribute to the progression of CRC through mechanisms such as DNA damage, oxidative stress, and inflammation specific to the colorectal environment.
Section 5:
Similar to the NF-κB and STAT3 pathways, the authors should clarify the specific roles of the PI3K/AKT and WNT/β-catenin pathways in different cell types within the tumor microenvironment, such as epithelial cells and immune cells. They should explain how these pathways contribute to the carcinogenesis process in CRC and their interactions within the TME.
Discussion:
The discussion can be significantly improved by discussing the role of inflammation in predicting cancer outcomes and highlighting the development and potential clinical application of anti-inflammatory treatments in managing CRC. For example, they should mention anti-inflammatory drugs, such as sulindac and aspirin, which are used to prevent and treat CRC by inhibiting NF-κB activity and reducing CRC risk.
When emphasizing the need for a multidisciplinary approach, the authors should reference specific studies that have successfully integrated GO and KEGG pathway analysis to understand the underlying mechanisms and clinical potential of the intestinal microbiome in CRC.
Conclusion:The authors should mention the role of the microbiome in CRC, highlighting its potential for therapeutic interventions.
Author Response
Dear Reviewer, first of all, thank you for taking the time to review our manuscript. We tried to address all your suggestions. Please find the answers to your comments in the following paragraphs:
Section 3.
The role of ROS and RNS in colorectal cancer carcinogenesis has been discussed.
Section 5.
The roles of the PI3K/AKT and WNT/b-catenin pathways have been clarified.
Discussion.
The clinical application of anti-inflammatory treatments in colorectal cancer has been mentioned, as well as specific studies of GO and KEGG pathway analysis have been referenced.
Conclusions.
The role of the microbiome in CRC has been mentioned, highlighting its potential for therapeutic interventions.